



# Have river flow droughts become more severe? A review of the evidence from the UK – a data-rich, temperate environment

Jamie Hannaford[1,2], Stephen Turner[1], Amulya Chevuturi[1], Wilson Chan[1], Lucy J. Barker[1],  Maliko Tanguy[1, 3], Simon Parry[1], Stuart Allen[4]

1.   UK Centre for Ecology & Hydrology, Maclean Building, Crowmarsh Gifford, Oxfordshire, OX108BB, UK
2.   Irish Climate Analysis and Research UnitS (ICARUS), Maynooth University, Maynooth, Co. Kildare, Ireland
3.   European Centre for Medium-Range Weather Forecasts (ECMWF), Reading, UK
4.   Environment Agency, Iceni House, IP3 9JD, Ipswich

**Correspondence:** Jamie Hannaford, jaha@ceh.ac.uk

**Abstract**

When extreme hydrological events (floods and droughts) occur, there is inevitably speculation that such events are a manifestation of anthropogenic global warming. The UK is generally held as a wet country, but recent drought events in the UK have led to growing concerns around droughts becoming more severe – for sound scientific reasons, given physical reasoning and projections for future. In this extended review, we ask whether such claims are reasonable for hydrological droughts in the UK, using a combination of literature review and extended analysis. The UK has a well-established monitoring programme and a very dense body of research to call on, and hence provides a good international case study for addressing this question. We firstly assess the evidence for changes in the well-gauged post-1960 period, before considering centennial scale changes using published reconstructions. We then seek to provide a synthesis of the state-of-the-art in our understanding of the drivers of change, both climatic and in terms of direct human disturbances to river catchments (e.g. changing patterns of water withdrawals, impoundments, land use changes). These latter impacts confound the identification of climate-driven changes, and yet human influences are themselves increasingly recognised as potential agents of changing drought regimes. We find little evidence of compelling changes towards worsening drought, apparently at odds with climate projections for the relatively near future and widely-held assumptions of the role of human disturbances in intensifying droughts. Scientifically, this is perhaps unsurprising (given uncertainties in future projections, the challenge of identifying signals in short, noisy records, and a lack of datasets to quantify human impacts) but it presents challenges to water managers and policymakers. We dissect some of the reasons for this apparent discrepancy and set out recommendations for guiding research and policy alike. While our focus is the UK, we envisage the themes within will resonate with the international community and we conclude with ways our findings are relevant more broadly, as well as how the UK can learn from the global community.





## 1. Introduction

Throughout much of 2022, the UK experienced one of the most severe droughts in recent decades (Barker et al. 2024). This episode followed a major drought in 2018 – 2019 (Turner et al. 2021) and this succession of events has naturally led to claims that such droughts are a manifestation of human-induced global warming, and that droughts have become more severe over time (e.g. Rivers Trusts, 2023). Such claims are entirely reasonable in that climate projections suggest droughts will become more severe in a warming world (e.g. in the latest eFLaG projections; Parry et al. 2024; for a more general summary see the review of Lane & Kay, 2023). These recent droughts have demonstrated the continuing vulnerability of the UK to drought, and underlined the need to understand whether and how drought risk is changing, and how it is likely to evolve in future.

A key aspect of understanding changing risk is in characterising past variability, to detect emerging trends and provide a baseline against which future changes can be quantified. In this extended review, we set out to capture the state-of-the-art in the evidence for *past variability in hydrological drought in the UK*, through a synthesis of the scientific literature complemented with additional new analyses to fill in several current gaps (Appendix A provides methodology for the extended analyses). This extended review is based on an earlier review conducted for the Environment Agency (Hannaford et al. 2023), compiled as part of a set of essays on the state of our knowledge on drought in the UK: Review of the research and scientific understanding of drought: summary report - GOV.UK (www.gov.uk). We also refer to several other essays throughout this paper.

Drought is widely written about as a complex, multi-faceted phenomenon that defies straightforward definition. Since Wilhite and Glantz (1985), drought has commonly been categorised into various types, often differentiating between meteorological, hydrological, agricultural droughts, alongside various others. This review focuses on *hydrological drought* (e.g. van Loon (2016)). More specifically, this review considers only *river flow* drought, and does not cover groundwater, lakes, reservoirs and so on. However, for convenience and brevity we use the term hydrological drought throughout.

Why are we interested in river flows? The simple answer is that river flows are one of the primary ways in which climate extremes (like droughts) have an impact on society and the environment, and through which climate change is likely to bring some of its most catastrophic consequences. Adequate river flows (of acceptable quantity and quality) are of fundamental importance to public water supply, abstractions for industry, energy and agriculture, for hydropower generation and for a host of other purposes including navigation and recreation. Moreover, river flows are vital for maintaining healthy aquatic ecosystems, and the many ecosystem services they support. Shortfalls in river flows during hydrological droughts can have





impacts for many economic sectors and cause increased competition between them, as well as between human
demands and the environment – with subsequent impacts on water, food and energy security in the long-term.
Additionally, river flows integrate across a range of processes occurring in a catchment. While many
meteorological measurements (notably, raingauges) sample only points in space, river flows represent the
combined balance of hydrological fluxes across large areas of the upstream land surface. River flows are,
therefore, a key broad-scale indicator of water availability, and long-term measurements of river flow enable
us to track hydro-climatic variability on a range of timescales. Nevertheless, due to the complicated processes
and timescales of drought propagation, from meteorological deficits to hydrological drought (van Loon et al.
2016; or Barker et al. 2016 for UK-specific context) it is necessary to quantify trends in river flows in
themselves rather than infer hydrological drought from precipitation or other climate variables.

This review is timely given growing recognition of drought as an important hazard in the UK. While the UK
is often thought of as a wet country, droughts are a recurrent feature (as in all climate zones) and, moreover,
some parts of southern and eastern England are relatively dry even by international standards. These areas are
already water stressed given significant socioeconomic demands (e.g. Folland et al. 2015) and in recent years
there has been growing concern about a future 'jaws of death' situation (Bevan, 2022) where demand outstrips
supply. Such fears have prompted major changes in water resource management, with water suppliers
challenged to ensure resilience to very extreme (1:200, 1:500 year) droughts, which has necessitated
significant innovations in planning techniques, alongside a growing trend towards regional- and national-scale
rather than local-scale drought and water resources planning (Counsell & Durant, 2023). Among the many
challenges of assessing resilience to such rare extremes, the question of non-stationarity of hydroclimate
variables like precipitation and river flows is an especially vexing one.

Hence, while this review is focused on the UK, many of the issues covered are of international import, and
will resonate in other hydroclimatic settings and governance frameworks. As this review will demonstrate,
there is a very dense literature on hydrological variability in the UK, and the UK provides an important
example for appraising change drought risk in a temperate setting where drought has historically been seen as
a relatively modest threat, in comparison to floods (e.g. Bryan et al. 2019; McEwen et al. 2022). We anticipate
that an accessible extended review will be of value for international comparisons and policymaking syntheses.
Despite countless publications on trends in drought or water resources variables, the evidence for consistent
trends in hydrological drought in international syntheses (including successive IPCC Reports) is
comparatively weak compared to other climate variables, largely due to deficiencies in available datasets
(Vicente-Serrano et al. 2022). The present review seeks to set out a comprehensive statement of evidence in a
data- and research-rich environment.

We will review the position of our knowledge of how droughts have changed by considering past trends and
variability in various river flow indicators relevant to water resources and drought (Section 2). This focuses on





the last five decades, the period of most UK river flow observations. We then take a longer view, looking at
river flow reconstructions over many decades back to the late 19th Century (Section 3). Importantly, we will
also consider the mechanisms (or drivers) behind variability in river flow drought. We address climatic drivers
(Section 4) and catchment drivers (section 5) – the latter encompassing changes in direct human interventions:
abstractions, discharges, reservoir management, land cover changes and so on.

The focus of our review is on investigating variability in river flows, and in particular river flow
characteristics relevant for drought (e.g. seasonal river flows, low flows), as well as indicators that are
designed specifically to characterise drought.  There is a substantial literature on the subject of drought
*indicators* and *drought indices* (e.g. WMO, 2016; Bachmair et al. 2016). We review studies that use a range of
drought indices that have been applied in the UK (e.g. the threshold level method, Rudd et al. 2017; the
Standardised Streamflow Index, e.g. Barker et al. 2016), and we apply these indicators in the extended
analysis presented here. For context, Fig 1 illustrates how drought indicators can be used to identify discrete
drought events, and quantify their characteristics (in terms of intensity, duration and accumulated deficit).

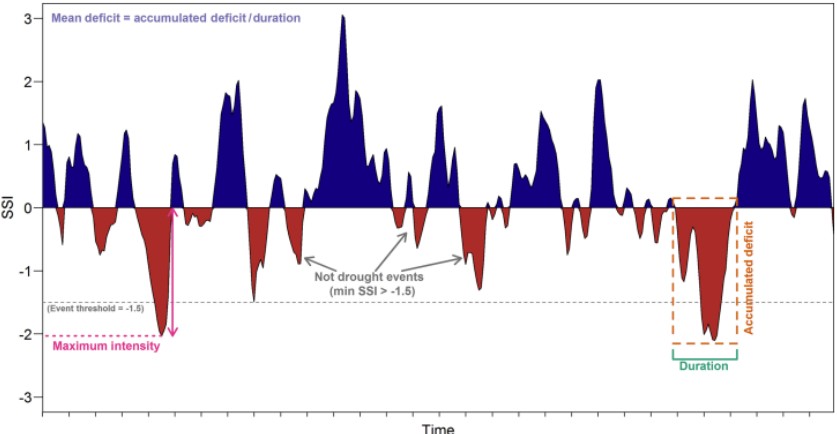


**Figure 1: conceptual diagram showing drought event characteristics when applied to droughts**
**extracted using a drought indicator (in this case, the Standardized Streamflow Index, SSI) applied to a**
**river flow time series. The SSI is a monthly time series, and droughts are defined as all events when the**
**SSI reaches a particular threshold (in this case, -1.5). The characteristics of the drought are then based**
**on the start (from when the SSI goes below zero) and end (when it returns above zero)**


**2.   Have hydrological droughts become more severe in observational records?**



In addressing the literature on past changes in drought, it is first important to highlight the very rich
information base on which assessments of past changes in hydrological drought is based. The UK has a very
dense hydrometric network in international terms, and is fortunate to have a centralised archive of accessible,
quality controlled hydrological data, the National River Flow Archive (NRFA; Dixon et al. 2013;
https://nrfa.ac.uk). This resource is the primary basis of most of the studies that have looked at past
hydrological variability highlighted in this section.
That said, there are inherent challenges in analysing long-term variability in river flows – as described in
Hannaford (2015), Wilby et al. (2017) and Slater et al. (2022). In particular, hydrological records are often
impacted by anthropogenic disturbances and constraints of poor data quality – particularly for extreme low
flows which are inherently challenging to monitor. This is especially important if trying to discern climate-
driven changes in river flow. In catchments with strong (or changing) levels of human disturbance, trends and
variations may not reflect climate variability. To this end, many countries have declared 'Reference
Hydrometric Networks' (RHNs) of near-natural catchments (Burn et al. 2012). The UK was an early leader in
this area, with the designation of the UK Benchmark Network (Bradford & Marsh, 2003; updated to UKBN2
by Harrigan et al. 2018). In the following sections, we contrast between some studies that use the Benchmark
network and those that apply to a wider range of observations from the NRFA.

A good starting point for any assessment of changing hydrological droughts are a series of previous 'Report
Card' reviews that addressed evidence for changes in river flows more generally (Hannaford et al. 2013, 2015;
Watts et al. 2013, 2015; see also update by Garner at el. 2017). These reviewed evidence for observed changes
in river flow across the UK (including both droughts and floods). These reviews summarised many studies
that analysed changes in variables such as annual flows, seasonal flows and low flows, with a very mixed
picture emerging as far as water resources/drought is concerned – at least compared to high flows/floods
where a more consistent picture emerged. Many studies are now quite old and covered data periods ending in
the 2010s. In general, there was limited evidence for any clear trend in annual low flows (e.g. Hannaford et al.
2006, based on data up to 2002). Low flow magnitude had typically increased (put another way, this indicates
less severe low river flows or droughts), particularly in the north and west. Seasonal flows showed increases
in winter and autumn, decreases in spring, and a very mixed picture in summer (e.g. Hannaford and Buys,
2012, based on data up to 2008). The Report Cards showed that there was little published evidence based
around changes in drought *per se*, using drought indices like threshold methods/Standardized Indicators, as
opposed to general flow regime indicators.

Since the publication of the Report Cards, there have been few additions to the literature on drought/water
resources trends. Harrigan et al. (2018) updated the Benchmark Network, and undertook an analysis of
seasonal trends and low flows, up to 2016, and found a very similar picture to previous assessments. Both
median (Q50) and low (Q95) flows showed increases in northern and western areas, but these were rarely



significant; decreases were observed across much of England, but these were typically non-significant and
there was substantial regional variation. Seasonal flows were consistent with past studies.

While there has been a recent update of flood trends (Hannaford et al. 2021) there has been no published
update of low flows or drought trends in parallel. For the purposes of this extended review, we have
undertaken a preliminary update of trends in low flows and seasonal flows, comparable with Harrigan et al.
(2018) but updated to September 2022 (the latest available data on the NRFA). This was done using the same
methodology outlined in Harrigan et al. (2018) and Hannaford et al. (2021) – see Appendix A. As with
Hannaford et al. (2021), we have deliberately compared the UK Benchmark Network (UKBN2) with the
wider whole-NRFA network. The time series end in September 2022, as the latest quality controlled NRFA
data and therefore does include the bulk and in most areas the 'peak' of the 2022 drought, despite in
continuing into October and beyond in some areas. While ending in a drought year could affect trends, a
previous version of this analysis excluding 2022 shows similar patterns (Hannaford et al. 2023).

For all the low flow indicators (Fig 2), the same general pattern emerges of increasing flows in northern and
western Britain, and a mixed pattern in the English lowlands. However, for the Benchmark network there is a
more recognisable tendency towards downward trends. For Q50 and Q70 there are few significant downward
trends, but more of the trends in northern Britain are increasing. For Q95, there are some significant
downward trends. Seasonal patterns (Fig 3) are similar to previous studies – generally, consistent increases in
autumn and winter, and decreases in spring, and a contrast for summer between increases in the north/west
and a mixed pattern, but with some significant decreases, in the south.  For spring and summer the patterns are
similar between the full network and UKBN2 sites, with spring showing decreases across the UK, and
summer showing increases in the north/west and decreases in the south/east. For autumn and winter, patterns
in the UKBN2 are more mixed, with both increases and decreases in England, although relatively few
significant; in Scotland however, all UKBN2 sites show increases.


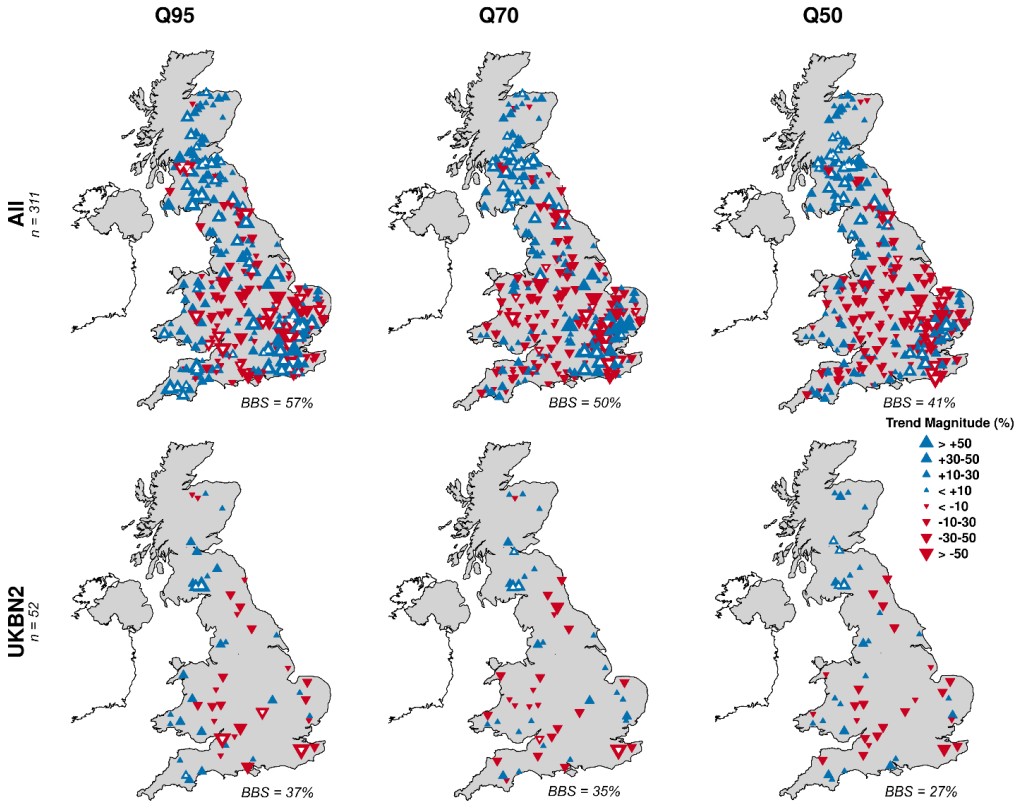

**Figure 2: trend analysis of river flow indicators relevant for water resources/drought (Q95, Q70, Q50) for the period 1965 - 2022. Top row = all NRFA catchments with available data (over this period). Bottom row = UK Benchmark Catchments suitable for Low Flow analysis. Trend magnitude is shown according to the key as a percentage change. White colouration of Triangles denotes a significant trend using the Mann-Kendall test (5% level), accounting for serial correlation where present. n.b. These are based on current NRFA data (to end of water year 2021-2022). The label 'n' denotes the number of catchments; BBS denotes the % for which a block bootstrap was used to account for serial correlation (see Appendix 1, methodology)**

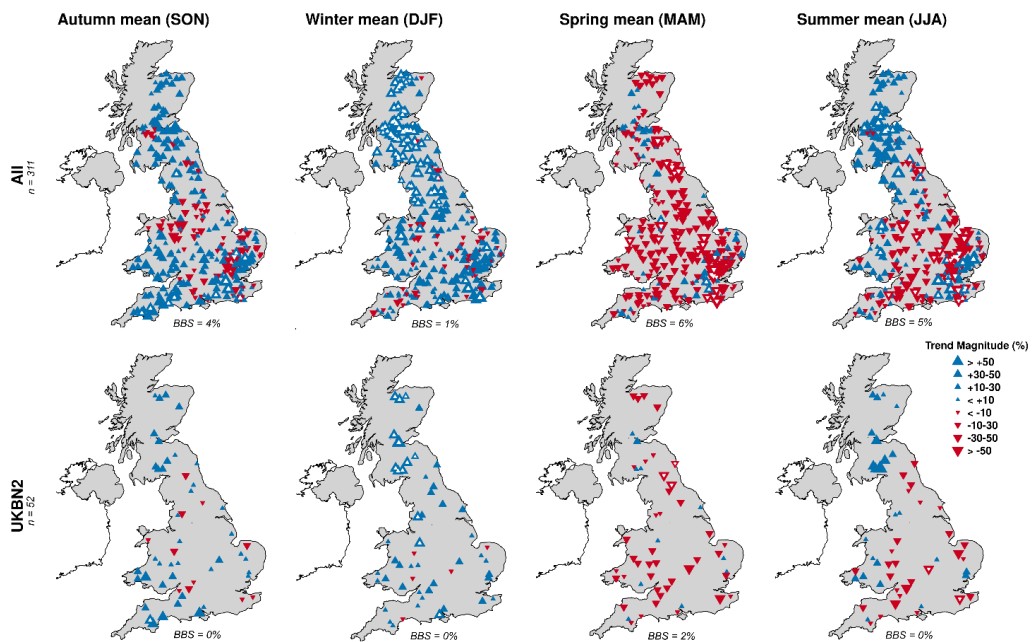

**Figure 3: trend analysis of seasonal mean river flows for the period 1965 – 2022 (see figure 2 caption for further explanation)**

It should of course be noted that past studies, and the above new analysis, are of broad indicators of 'drought relevant' seasonal and low flows, rather than analysis of droughts *per se*, using the kind of indicators highlighted in the introduction. Such studies have not previously been published in detail at the UK scale although Pena-Angulo et al. (2022) analysed hydrological drought trends between 1962 and 2017 using the SSI, at a European scale, and included 474 UK catchments in their study, embracing a range of both natural and influenced catchments. They found largely negative trends in drought frequency, duration and severity (i.e. towards fewer, shorter and less severe droughts) for the UK, albeit also with very mixed patterns. Significant trends towards an amelioration of drought severity were more prevalent in northern and western catchments.

Here, we have conducted a similar analysis for the UK Benchmark network, using droughts extracted using drought indicators (Fig 4). We show results for the SSI3, but similar analysis using threshold level methods is shown in A1. Very similar results to Pena-Angulo et al. (2022) are found, with trends towards decreasing drought severity in the north and west and a mixed pattern in the southeast, although with some spatially coherent (but rarely statistically significant) trends towards worsening drought.


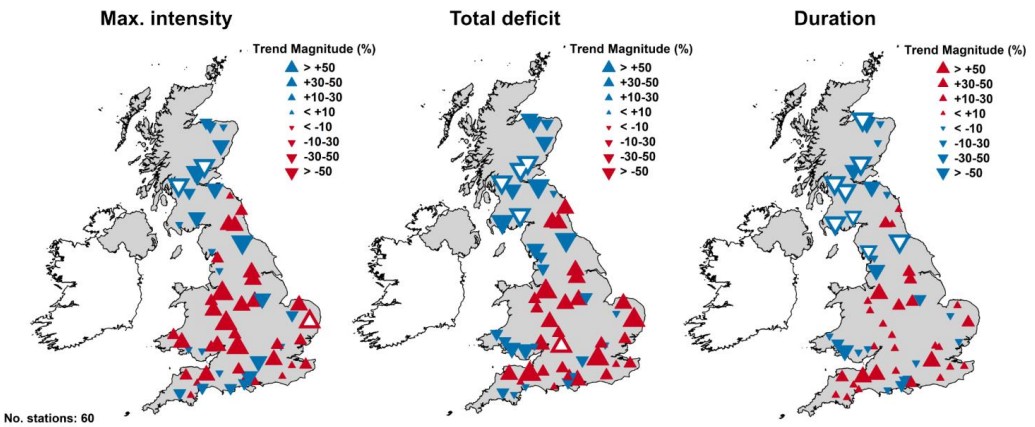



**Figure 4: trend analysis of extracted hydrological drought characteristics using SSI3 for the LFBN for**
**the period 1965 – 2021 (see figure 2 caption for further explanation). Note the different scales used for**
**each: for intensity and deficit, positive trends mean decreasing drought whereas for duration, positive**
**trends mean increasing severity. Hence, for ease of interpretation, in all cases red signifies worsening**
**drought and blue amelioration of drought**


It is important to underscore that observed trends are very sensitive to the period of analysis. The new results
presented here in Figs 2 - 4, alongside previous studies, typically analyse linear, monotonic trends in a fixed
period. Other studies have adopted a 'multitemporal analysis' to look at sensitivity of trends to start and end
point, and find that varying the start or end by even a few years can radically change the outcomes, with
changes in significance and even the direction of change. Hannaford et al. (2021) demonstrate this for flood
trends for the UK, but a similar comprehensive analysis of sensitivity to low flow or drought trends is lacking
in the published literature. Wilby (2006) and Hannaford & Buys (2012) showed how varying start years
influenced annual, seasonal and low flow trends. In general, trends over the typical 'observational' period
(post-1960s) are often somewhat different to those seen in longer hydrological records. The increases in
summer and low flows seen in many published studies partly reflect the fact that the late 1960s to mid-1970s
was notably dry, and the late 1990s – late 2000s was generally much wetter. Murphy et al. (2013) highlight
how positive trends are consequently 'locked in' by the coverage of typical gauged records in Ireland, and the
UK picture is very similar. This underscores the importance of taking a longer view than the typical gauging
station record length, as discussed in Section 3, where we extend the window of analysis and examine
multitemporal trends in drought.




### 3. Historical hydrological droughts – a long view using reconstructions


Recent droughts have inevitably invited comparisons with past drought events (e.g. Parry et al. 2022, Turner
et al. 2021) and these have shown that 2022 and 2018 droughts rank among some of the most significant
hydrological droughts of the last 50-years in terms of low flows. Previous drought events of the 2000s and
1990s were also extensively documented at the time (e.g. 2010 – 2012, Kendon et al. 2013; 2004 – 2006,
Marsh et al. 2007) and again, these events were found to be significant in the context of the typical gauged
record – that is, from the 1960s/1970s, when the majority of UK gauging stations were installed.

Despite the half-century coverage of many gauging stations, which is impressive in an international context,
the 'instrumental' record only contains a handful of major drought events. To appraise drought risk more
fully, many authors have highlighted the need to examine droughts over much longer timescales. This is
important for water resources management, particularly in the context of the deep uncertainty in future climate
projections. While the past may not be so readily a guide to the future in a warming world, at the same time
observed historical droughts represent an important benchmark of drought risk, given that these events have
actually unfolded – they also offer the opportunity to learn from past experiences in drought management.
Historical droughts have, therefore, always formed a cornerstone of water resource planning. While recent
developments have moved away from a single 'drought of record', i.e. a worst drought used as a stress test, to
considering droughts more severe than the observed envelope (using stochastic methods and other
approaches) (e.g. Counsell & Durant, 2023), these methods are ultimately still dependent on past observations.
A fuller understanding of historical hydrological droughts is therefore of critical importance to practitioners.

The influential study of Marsh et al (2007) identified major droughts in England and Wales back to 1800. This
study highlighted the prevalence of major drought events in the pre-1960 era, and underlined the importance
of events such as those of the 1920s, 1930s and the 'long drought' period spanning the turn of the 20th century,
as well as some droughts in the 1800s which are relatively poorly understood. Marsh et al. 2007 considered
drought primarily from a meteorological perspective, given the abundance of long rainfall records – although
these authors did gather hydrological evidence, where available, and moreover documented evidence of
impact of past drought episodes. From a hydrological viewpoint, such comparisons are challenging given that
very few gauging stations captured the droughts of the 1920s – 1940s or earlier.

To fill this gap, there have been several efforts to extend hydrological records through reconstruction,
primarily using rainfall-runoff models to estimate past river flows given the long meteorological records
available as input. The earliest work of Jones (1984) was updated by Jones et al. (1998) and Jones et al.
(2006), and delivered monthly reconstructions (hereafter, CRU reconstructions) back to 1860 for 15
catchments in England and Wales using a simple statistical water balance model driven by long raingauge
series. Jones et al. (1998) used a 'Drought Severity Index' (DSI) to identify major droughts in these records,



and highlighted that in no cases were the contemporary droughts of the 1970s – 1990s the most severe
droughts in the longer-term records.

More recently, as part of the 'Historic Droughts' project, Smith et al. (2019) delivered a dataset of
reconstructed river flows for 303 UK catchments (Historic Droughts reconstructions) using the GR4J
hydrological model, driven by a newly-updated high-resolution daily gridded precipitation dataset and
Potential Evaporation (PE) reconstructed from gridded temperature (using the approach of Tanguy et al.
2018). Barker et al. (2019) then used these reconstructions to conduct an analysis of historical hydrological
droughts and their relative duration and severity using the SSI, for 108 benchmark catchments (Figure 5). In
common with previous studies, these authors showed that while recent droughts in the well-gauged era (post-
1960) rank highly, there are many historical episodes that are longer or more severe than those of the recent
past. A separate reconstruction was conducted for the 'MaRIUS' project by Rudd et al. (2017) using a
distributed model, Grid2Grid, also driven by gridded meteorological inputs, and with droughts extracted using
a fixed threshold approach. Barker et al. (2019) and Rudd et al. (2017) found, unsurprisingly, good agreement
with the droughts identified by Marsh et al (2007). However, these studies highlight important departures, e.g.
the importance of droughts in the 1940s that are not well-attested in impact terms due to wartime reporting
(Dayrell et al. 2022) and the late 1960s and early 1970s – the impacts of which were eclipsed by the 1976
event. Importantly, both Rudd et al. (2017) and Barker et al. (2019) concluded that there were no obvious,
discernible trends in hydrological drought (cf. Fig 5) in these centennial scale reconstructions. However, no
formal trend tests were carried out.

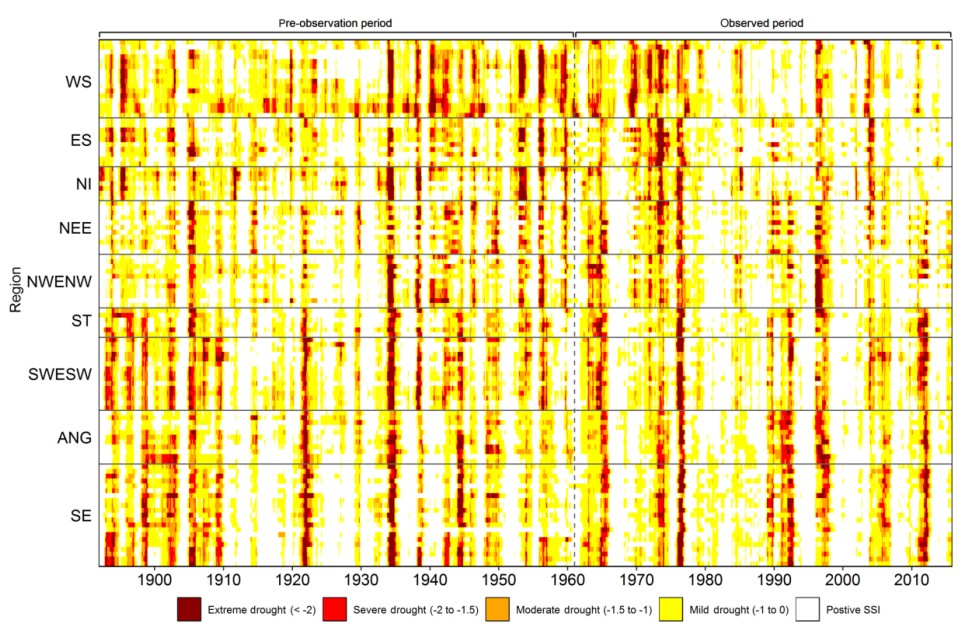






**Figure 5 – Heat map of SSI-12 for LFBN catchments from 1891 to 2015 (catchments arranged roughly**
**from north to south on the y axis, with one row per catchment and hydro-climatic regions marked for**
**clarity) with colours according to SSI12 category in key. 'Observed period' highlights typical maximum**
**record coverage of most gauging stations. 'pre-observation' the period with most added value from the**
**reconstructions. Reproduced with permission from Barker et al. 2019**

Here, we augment previous work by examining drought trends using multitemporal analyses (after Hannaford
et al. 2013, 2019; see Appendix A) applied to the reconstructions of Barker et al. 2019 for a selection of
catchments (the same nine appearing in that paper, giving a good geographical spread across the UK). The
results (fig 6) show very strong sensitivity to the period of analysis. In the north and west (Cree, Allan Water,
Ellen) there is generally a contrast between decreasing drought severity in drought when analyses start from
the mid-20[th] century and end in the present, whereas earlier start periods show trends towards increasing
severity. Very few periods show statistical significance. In other parts of the country there are more mixed
variations. The Lud and Lambourn show a greater propensity towards increasing severity, with the Lud
showing more recent start dates and the Lambourn showing the reverse. For the Lambourn, interestingly,
positive trends emerge when analyses begin pre-1910 (the 'Long Drought' was especially significant in
groundwater catchments in the southeast). Overall, however, while interesting contrasts can be drawn,
statistically significant trends are rare – these selected reconstructions confirm the assertion of Barker et al.
(2019) that here is little evidence for consistent patterns towards worsening drought over the long-term.



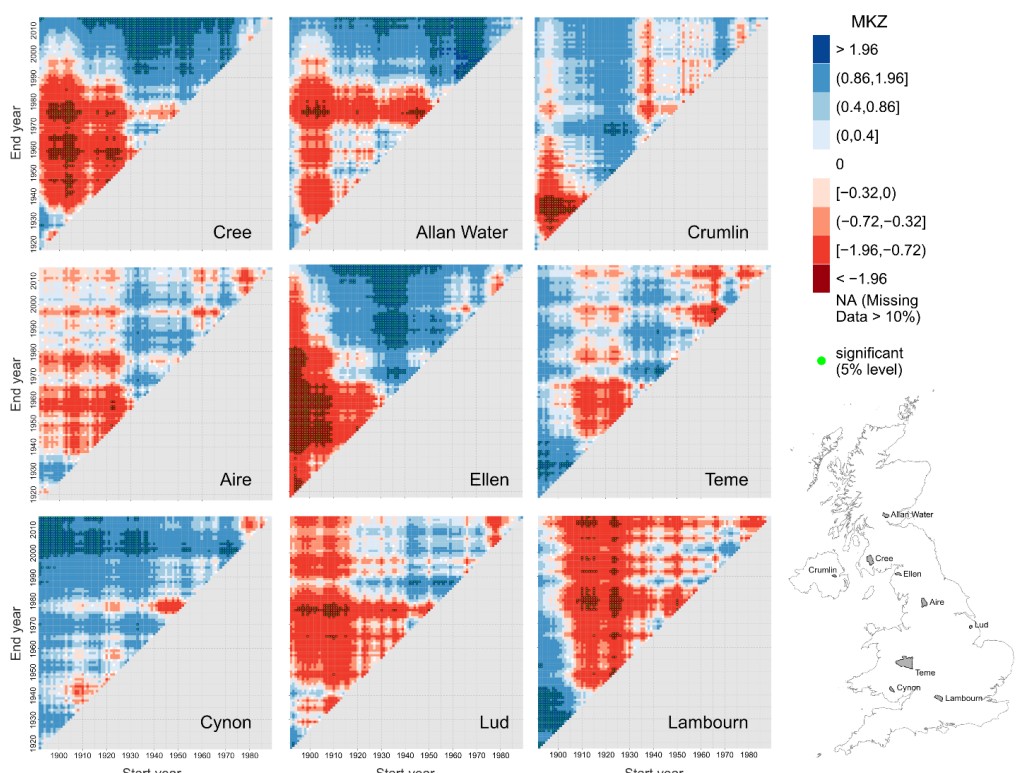

**Figure 6: Multitemporal trend analysis applied to time series of accumulated drought deficit using SSI3 for nine selected long reconstructed records from Barker et al. 2019. The colour ramp denotes values of the MK Z statistic (blue = positive, red = negative) with green dots denoting significant cases.**

Following on from this theme of identifying 'droughts of record' for water resources planning, several other noteworthy studies have reconstructed hydrological droughts on a regional basis, and then fed these into water supply system models, e.g. for East Anglia (Spraggs et al. 2015) and the Midlands (Lennard et al. 2015). Interestingly, in both cases it was found that an extended reconstruction of droughts into the 19th Century made little difference to water supply yields – that is, the additional 19th century droughts did not test water supply systems more than those in the available long rainfall records (generally, back to the 1920s). However, these conclusions are regional and system-specific, so further research is needed to see if the Historic Droughts/MaRIUS reconstructed hydrological droughts make a significant difference in other parts of the country.




### 4.    Drivers of change in hydrological drought – climate factors


Trends and past variations in river flows such as those described in section 2 and 3 can be driven by either
climate or non-climate (catchment) factors. Some effort to isolate the climate-driven signal has been made
through the identification of Benchmark catchments. However, having established a 'control' network for
detecting climate-driven changes, the question remains of what mechanism is behind the observed river flow
change. Most pertinently, the question is whether observed changes are attributable to anthropogenic
warming, or due to variability in the wide range of natural, internally forced modes of ocean-atmosphere
variability. More realistically given the extent to which these factors are intrinsically linked, the answer is
'some combination of both', and the question is whether the relative roles can be disentangled and quantified.
This is not an abstract question, as the time evolution of future trends will depend on the balance between
'thermodynamic' anthropogenic warming, which is unidirectional to all intents and purposes, and circulation-
driven changes which could amplify, moderate or even counter such trends.

In this section, we briefly review the literature on the hydroclimatology of UK droughts, i.e. on climate-river
flows associations, to understand what climate factors have been linked to variations in UK river flows.
Knowledge of this topic is central to the climate detection and attribution debate, and yet is also of practical
importance for the development of monitoring and seasonal forecasting systems.

Firstly, we can compare river flow trends with published studies of basic meteorological variables relevant to
water balance (precipitation, evapotranspiration). River flow trends are consistent with observed climate
trends, notably significant trends towards wetter winters and, to a lesser extent, autumns, and a pronounced
spring drying in the recent past (Kendon et al. 2022). Other studies have also found significant increases in
evapotranspiration in spring (Blyth et al. 2019), in addition to spring drying. Summers have, in general,
become wetter over the same period as that featured in most river flow studies, but there has been a period of
generally wetter summers since c.2007, and drier summers in the 20-30 years before (Kendon et al. 2022). In
general, though, river flow trends (Figs 2 – 4) like meteorological analyses, shows little compelling evidence
(beyond a few catchments with significant downward trends) for any pronounced decreases in summer, nor
for low river flows – i.e. the kind of water availability indicators most relevant for drought. This is somewhat
at odds with future projections which consistently suggest substantial decreases in summer rainfall, flow, low
flows, and associated increases in drought severity (e.g. summarised in Lane & Kay, 2023) for the relatively
near future. We return to this in our discussion below.

We next consider the most extensively studied climate-hydrological associations – those connections, or
teleconnections, between river flows and larger-scale, lower frequency modes of variability – atmospheric
circulation indices such as the North Atlantic Oscillation (NAO). The NAO is the leading mode of variability





in the euro-Atlantic sector, and as such is an obvious candidate for linking with river flows. The NAO,
through its strong control of the location of the storm track and thus moisture delivery to the British Isles, has
long been shown to strongly influence UK rainfall, especially in the winter months, and it follows that river
flow patterns can also be linked to NAO variability. There is a large literature on this topic which we will not
cover in detail here. But this literature is consistent in showing very similar patterns, namely a strong positive
association between the NAO Index (NAOI) and river flow in the winter months, especially in northern and
western areas. However, relationships are complex, especially in non-winter months, and especially in the
lowlands of southern and eastern England, where the effect of the NAO is modest and, again, strongly
catchment-controlled (e.g. Laize and Hannah, 2010; West et al. 2021). The NAO is not the only relevant
pattern, and other studies have shown a prominent role of other modes of variability (notably the East Atlantic
pattern and the Scandinavia pattern, e.g. Hannaford et al. 2011; West et al. 2022). West et al. (2022) linked
NAO and EA patterns to the SPI and SSI, and highlighted the interaction of these modes of variability,
throughout the year, and note how their relative role varies around the country as well as seasonally – as well
as the role of propagation from SPI to SSI. While the NAO dominates in winter in the north and west, it has
far less explanatory power in the south and east in summer, when the EA plays a key role in modulating the
NAO influence.

The upshot of the strong control of the NAO, EA and other modes of variability is that the time evolution of
river flows, and drought indicators to an extent, can be seen to be controlled by the variability and interplay of
these patterns. A prominent role for the NAO has been claimed for explaining trends towards wetter winters
(and higher river flows) in northern and western UK (e.g. Hannaford et al. 2015, and references therein) over
the 1960s – late 1990s especially when the NAO was primarily positive. However, since then the NAOI has
been more variable yet trends towards higher winter flows have been unabated. The picture is a very complex
one, and recent studies have shown strong non-stationarity in the relationship between the NAO and UK
rainfall and river flows (as well as groundwater levels) over long timescales (e.g. Rust et al. 2022).

While the dipole-based NAO, EA, SCA and synoptic scale drivers can explain some variability of
hydrological drought occurrence, there is arguably even greater benefit from zooming out still further to
consider the role of larger-scale, slowly varying ocean-atmosphere drivers - notably (quasi-) cyclical patterns
of sea-surface temperature variations such as El Nino-Southern Oscillation (ENSO) or the Atlantic
Multidecadal Oscillation (AMO) that themselves influence the state of the NAO. Such patterns have a
reasonable degree of predictability, so uncovering robust links between them and river flow could have
profound implications for efforts to forecast and project water availability. Folland et al. (2015) reviewed the
state of knowledge of such links at the time, and demonstrated links between ENSO, and a range of other
predictors, and UK (specifically, lowland England) rainfall – most notably with La Niña events (links which
have been long established; see references therein). They also showed the impacts of La Niña on river flows
and groundwater, including drought indicators like the SPI/SSI for the Thames region. While links between



La Niña events and English lowlands winter half-year droughts were uncovered, such relationships are weak
and highly non-linear.

More recently, Svensson and Hannaford (2019) also took a global scale approach to explore links between UK
regional rainfall and river flows on the one hand, and SST patterns in both the Atlantic and Pacific oceans.
These authors confirmed an impact of Pacific Ocean variability (the Pacific Decadal Oscillation, strongly
linked to ENSO), but found it was highly modulated by the state of the North Atlantic (Figure 7). Such
relationships were present not just for the winter, but in summer months, previously considered much less
promising for forecasting, and yet of the most importance for drought management. The implication is that to
understand UK river flow variability, and hydrological drought, it is necessary to look well beyond WTs or
even dipole-like circulation indices, and zoom out to take a global view of atmosphere-ocean dynamics.

To identify regions significantly influencing UK droughts beyond the North Atlantic, we applied
methodologies similar to those used by Svensson and Hannaford (2019). The impact of remote climate drivers
was analysed across three distinct UK regions with varying SSI catchment characteristics: the north-west, a
transition zone, and the south-east (Figure 7a). We performed regressions of the area-averaged regional
Standardized Streamflow Index (SSI) time series for these regions against the global SST dataset at each grid
point, both concurrently (Figure 7b-d) and with a six-month lag (Figure 7e-g). See Appendix Section 3 for
more details on the data and methods used.

As expected, our results highlight the North Atlantic as a significant driver for all the three regions of UK
(Figure 7b-d). Additionally, the equatorial Pacific Ocean has strong correlations with SSI in all three regions
of UK concurrently and with a lag of 6 months. Indian Ocean shows significant correlations concurrently with
all UK SSI (Figure 7b-d), but at a lag of 6 months Indian Ocean influence is associated with only south-east
UK (Figure 7g). Similarly, southern Atlantic Ocean only has strong correlations with south-east UK (Figure
7d,g).



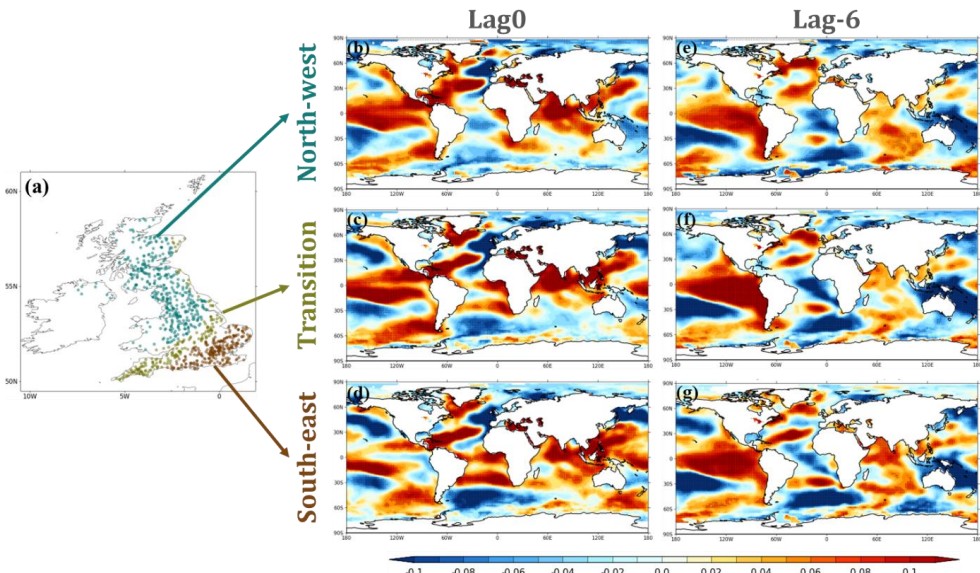



**Figure 7: (a) Three distinct regional clusters for catchments based on SSI (north-west, blue; transition, green; south-east, brown) identified using the 3-month accumulation of SSI timeseries for 1960-2020 using k-means clustering. Regression (shaded) between each grid point of SST and SSI for (b) north-west UK at lag0, (c) transition region of UK at lag0 and (d) south-east UK at lag 0, with regions significant 0.05 level demarcated with stippling. (e-g) same as (b-d) but with lag-6 months (i.e., SST lagging by 6 months to SSI).**

461

Despite the strong linear relationships between the Pacific, Atlantic and Indian Oceans and the UK climate, these teleconnections might not be direct, linear, or even stationary (e.g., as noted for Pacific influences by Lee et al., 2019). Multiple pathways have been proposed for these teleconnections, linking distant regional SSTs to the North Atlantic, which will ultimately influence UK hydrology. The Tropical Pacific's influence on the North Atlantic-European region has been identified through: (i) the stratospheric pathway leading to sudden stratospheric warming via the polar vortex (e.g., Trascasa-Castro et al., 2019), (ii) the shifted Pacific jet associated with transient eddies entering the Atlantic region (Li and Lau, 2012), and (iii) the Rossby wave train affecting the Pacific–North America sector (Mezzina et al., 2020). In the context of droughts, Tropical Pacific variability may shift the North Atlantic jet (e.g., Madonna et al., 2019) or cause blocking high pressures over the European region (e.g., Cassou et al., 2004), leading to severe droughts and heatwaves across Europe. Studies have also found that warming in the Tropical Indian Ocean leads to changes in the North Atlantic through a positive NAO-like response, which explains the development of the North Atlantic "warming hole" (Hu and Fedorov, 2020), or through the strengthening of the Atlantic meridional overturning circulation (Hu and Fedorov, 2019). Additionally, there are pathways that combine the influences of the





Indian Ocean Dipole and El Niño-Southern Oscillation (ENSO) on the North Atlantic Oscillation (Abid et al.,
477   2023).


In general, there have been some advances in explaining the drivers of hydrological drought through relating
various climate/ocean indices to river flow indicators. Fewer studies, however, have linked to drought
indicators specifically.  In addition, while such relationships have been used to explain observed river flow
variability and trends, most have been what may be termed 'soft attribution' through associations and
correlation. There have been few 'hard attribution' studies (Merz et al. 2012), that is, studies that have
demonstrated conclusively a causal chain between climate variations and trends in river flow ('proof of
consistency', Merz et al. 2012) and also ruled out other factors (proof of inconsistency) – e.g. catchment
changes, as discussed in section 5.

A second aspect of attribution is separating any signal of anthropogenic warming from internally-forced
variations such as ENSO, AMO and so on, discussed above. Formal climate detection and attribution studies
have been undertaken for UK flood events (e.g. for the 2013-2014 floods; Schaller et al. 2016). Attribution
studies for drought are less common, at least those that focus on the UK specifically, but the role of human-
induced warming has been shown for the wider European 2022 meteorological drought (e.g. Faranda et al.
2023). More generally, detection and attribution studies have been undertaken for meteorological drought
globally (e.g. Chiang et al. 2021), but they have not been applied for hydrological indicators. A majority are
also event-based rather than attributing long-term trends. Gudmundsson et al. (2021) claimed global trends in
mean and low river flows could be attributed to climate warming, but ideally such studies need replicating at
the finer scales relevant for UK water management policymaking and practice.


**5. Drivers of change in hydrological drought – human factors**
As shown in Section 4, there is a substantial and growing literature on the links between climate drivers and
hydrological drought, motivated by the need to understand the factors controlling large-scale water
availability. In many UK catchments (in common with many other domains, globally), however, river flows
patterns often deviate markedly from climate variability due to pervasive artificial influences on river flow
regimes. While RHNs enable climate signals to be discerned, many RHN sites are small, headwater
catchments in the uplands, and are often some distance away from major population centres. Arguably the
most important locations are those in the heavily populated, intensively managed lower reaches, where
understanding climate and human controls on hydrological drought is much more challenging.
Hence, while RHNs seek to filter out artificial influences as a 'control', these influences are worthy of study
in and of themselves. This has been the spirit of the International Association of Hydrological Sciences
(IAHS) 'Panta Rhei' decade (https://iahs.info/Commissions--W-Groups/Working-Groups/Panta-Rhei) that has
sought to understand and quantify human influences on flow regimes, and that has spawned a 'drought in the





Anthropocene' initiative (van Loon et al. 2016). Internationally, many studies have attempted to quantify the
impact of influences such as reservoirs, abstractions, discharges and other regulation on flow regimes and,
thence, on drought characteristics (for example, see the overview of van Loon et al. 2022). Such surveys
highlight the many challenges in discerning the impact of any particular human influence because multiple
impacts occur in parallel, are difficult to disentangle and may offset or compensate for one another.
Nevertheless, in spite of these challenges, these are not just academic debates, but topics of huge societal
import: in the UK, there is a long-standing, and sometimes polarised and contentious, debate on the role of
abstractions on hydrological drought and low flows, especially for Chalk streams, that has attained particular
prominence in recent years (e.g. CaBa, 2021).

Despite this growing interest, in both academia and the public eye, there have been relatively few UK studies
in the scientific literature that have conclusively linked artificial influences (or, commonly, a change in
artificial influences) with hydrological drought responses. Partly, this reflects the challenges of obtaining
suitable datasets of artificial influences. In the absence of directly available datasets of influences, researchers
have resorted to indirect techniques. Tijdeman et al. (2018) took a 'large-sample' approach to compare the
drought regimes of catchments classified according to the presence/absence of certain influences, using the
NRFA's Factors Affecting Runoff or FAR codes. While the study suggested that deviations in drought regime
(i.e. expected response to precipitation) could be linked to influences (notably, extended drought durations
linked to the presence of groundwater abstractions in Chalk catchments; Fig 8), in practice the method was
primarily a screening approach, and no quantitative proof could be offered in the absence of data on impacts.
Bloomfield et al. (2021) also took a large sample approach, using the CAMELS-GB dataset, which does
incorporate some limited artificial influences data within, to develop statistical models to assess the impact of
abstractions, discharges and reservoir operations on baseflow in 429 catchments. Inclusion of such water
management interventions improved the statistical models in some cases – especially for groundwater
abstractions, suggesting a detectable impact, in common with Tijdeman et al. 2018. These authors note that
more detailed information on water management than is currently available in CAMELS-GB would be needed
to fully constrain the specific effects of individual water management interventions on Baseflow Index (BFI).
More recently, Coxon et al. (2024) applied Machine Learning approaches to CAMELS-GB, and highlighted
the role of wastewater discharges in dominating low flow signals in urban catchments. This study was not able
to show *changes* in discharge inputs influencing changing low flow or drought properties over time, given the
static nature of the information on human impacts – but given the pervasive nature of such impacts
demonstrated, it is easy to see how catchments experiencing changes in abstractions, discharges or the balance
between them could see changing drought or low flow regimes.

Salwey et al. (2023) took a large sample approach to detect reservoir impacts on river flows using
hydrological signatures, including low flow metrics. They compared signatures from 111 Benchmark
catchments with 186 catchments modified by reservoirs. They found that reservoirs create deficits in the water



balance and alter seasonal flow patterns, while low flow variability was dampened by reservoir operations.
This approach of comparing signatures between Benchmark and impacted datasets enabled identification of
thresholds above which the reservoir 'signal' could be isolated from wider hydroclimate variability, and holds
promise for discerning the effect of other human impacts.

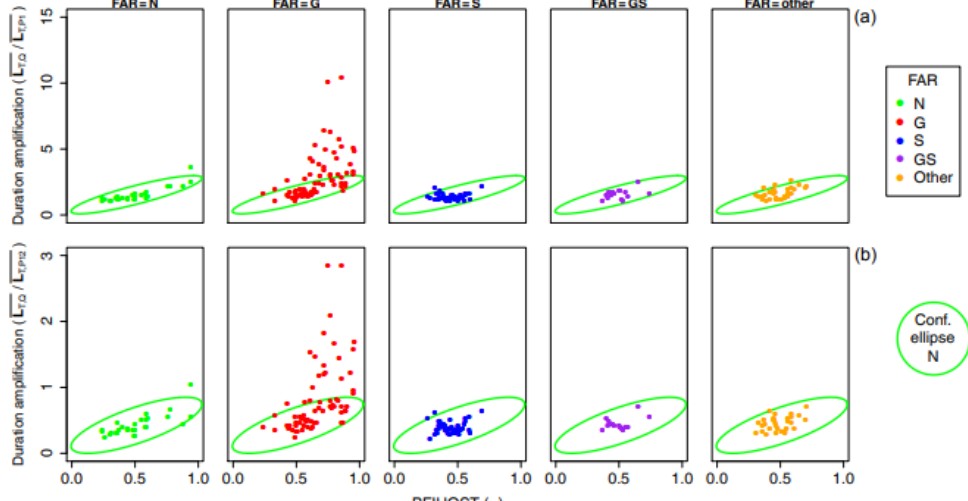


**Figure 8: amplification of average monthly streamflow drought duration over average monthly**
**precipitation drought duration (top: 1 month precipitation; bottom: 12 month precipitation) versus**
**BFIHOST for catchments labelled with different 'Factors Affecting Runoff' codes (colours). Ellipse**
**reflects the 95 % confidence ellipse for catchments with near-natural flow records (FAR = N). FAR = G**
**(groundwater abstraction) shows many catchments have longer droughts than expected based on**
**precipitation. Reproduced with permission from Tijdeman et al. (2018)**

Other studies have adopted paired catchment analyses – e.g. van Loon et al. (2019) who compared droughts in
two hydrologically-similar catchments in eastern England, with one catchment impacted by a water transfer
scheme, while Coxon et al. (2024) also used paired catchments to demonstrate the role of wastewater
discharges on flow regimes. While differences can be observed in drought characteristics, once again there is
limited or no time-varying information on the human influences (abstractions, discharges) to prove the effect
conclusively ('weak attribution' in the parlance of Merz et al. 2012).

It follows that there are few studies that show a change or trend in UK river flows, or relevant drought
indicators, that can be attributed to artificial influences, beyond the observation of Tijdeman et al. (2018) of a
*tendency* towards increased drought anomalies over time in many catchments affected by groundwater
abstraction. The reverse may also apply when abstraction decreases. Clayton et al. (2008) noted an increase in
river flows since the cessation of a major groundwater abstraction in the river Ver, as part of an alleviation of





low flow (ALF) scheme, but again noted this could not be confidently attributed to that cause alone. Similarly,
Tijdeman et al. (2018) show a similar example for the Darent, a river with an ALF scheme, although also
conclude that such relationships need further work to fully elucidate.

While the literature on artificial influence impacts on drought is relatively sparse, the situation is even more
acute for the influences of land use or land cover (LULC) change, despite this being a long-standing topic in
UK (and global) hydrology. This is certainly the case for low flow and drought indicators, that have arguably
been neglected in comparison to floods, for which there have been many studies. Nevertheless, reviews and
meta-analyses show that there is very limited consensus on the extent to which flood indicators are
conclusively influenced by rural land management (e.g. O'Connell et al. 2007), afforestation (Stratford et al.
2017) or Natural Flood Management (Dadson et al. 2017). For water resources or drought indicators, there
have been no major efforts to synthesise the literature in a comparable way.

At the catchment scale LULC have been very comprehensively investigated, for isolated catchments – with
the most notable example being the paired catchment studies at Plynlimon, mid-Wales (see the review of
Robinson & Rodda, 2013). The Plynlimon experiment did not investigate drought responses *per se*, but
showed the impact of afforestation on catchment evaporative losses and, hence, river regimes, including low
flows. While there has been growing interest in quantifying the effect of afforestation on flood regimes, as a
potential mitigation strategy, there have been few studies looking at drought or low flows at the larger scale.
Recently, Buechel et al. (2022) used (land cover) scenarios of potential afforestation applied to a land-surface
model (JULES) to quantify the effect of afforestation in twelve diverse (and generally large) UK catchments.
Surprisingly, given vigorous debates on the topic, these authors found little impact on flooding, but much
larger impacts at median and low flows. It must be noted this was a scenario-based ('what if' scenarios) rather
than observational study.

Urbanisation is a major potential impact on streamflow regimes, but again the focus has largely been on
investigating the effect of urbanisation on flood frequency (e.g. Prosdocimi et al. 2017). Few studies have
investigated wider streamflow regimes more generally. However, in an interesting development, a recent
study by Han et al. (2022) investigated non-stationarity in observed river flow regimes in twelve urbanising
catchments (using datasets of changing urban cover) and found that the strongest signals to emerge were for
low flows rather than high flows. While increases in urbanisation tended to increase the magnitude of flows
across the whole regime, the rate of increase was much higher for low flows (1.9% ± 2.8%(1s.d.) for every
1% or urban cover) than high flows (0.5%±2.2(1s.d.)).

In summary, the impact of human interventions on hydrological drought rests on a very limited evidence base.
One major limitation has been the availability of impact datasets. There have been significant advances in
developing datasets of impacts of abstractions and discharges for England, based on the Environment





Agency's data holdings – notably CAMELS-GB (Coxon et al. 2020, 2024) and the gridded dataset of
Rameshwaran et al. (2021).  Barriers remain to access of underlying abstractions and discharges, but these
derived products are important community assets and further studies will no doubt emerge using them.

**6.   Discussion and recommendations for future directions**
When there are major drought events, it is often said that droughts are becoming more severe due to
anthropogenic warming. While the evidence for human warming is unequivocal, it cannot be said so readily
that there is compelling evidence for changes in hydrological drought in the UK – certainly there is not (yet)
strong evidence for droughts becoming more severe despite the occurrence of two major hydrological
droughts in the last half-decade. In contrast, there are sound scientific reasons why we should expect changes
to hydrological drought in a warming world, and future projections indicate we will (Lane et al. 2023).
Clearly, reconciling past observations and future projections remains as big a scientific challenge as was
highlighted in past reviews (Hannaford 2015; Watts et al. 2015).

This lack of congruency between historical observations and future projections has been called a 'conceptual
controversy' in the past by Wilby et al. 2008. That study referred to floods, and arguably the gap between
projections and observations has narrowed significantly in the recent past for floods – but while there is
increasing confidence in studies detecting fluvial flood trends, this is not the case for hydrological drought.
However, as argued in the original paper (Wilby et al. 2008) it is important not to see 'controversy' as a
reason for inaction. There are good reasons why the disparity emerges: projections inevitably span a large
range of uncertainty; with observations, signals are weak and obscured by natural variability, as well as by the
impact of direct human disturbances. The lack of compelling trends in drought or low flows can be seen by
the sensitivity to study period, and how readily strength or directionality of trends changes with small shifts in
perspective. This arises because of strong interannual and interdecadal variability due to a range of large-scale
atmospheric/oceanic circulation patterns (see Section 4). Wilby (2006) highlighted that it can take very long
'detection times' of many decades for a signal of anthropogenic warming to be detectable above the noise of
interannual and interdecadal variability. In this context it is unsurprising that 'detectable' (i.e. statistically
significant) trends may not yet have emerged, even if there is an underlying anthropogenic component. Wilby
(2006) argues that trends may be *practically* significant for water managers way before they become
statistically significant.

Looking across this synthesis, we can conclude that while there are some gaps, a comprehensive body of work
exists on past variability in UK drought. Given this fact, a conclusion that highlights relatively little evidence
for change, contrary to near-future expectations, may seem surprising. Our question was 'have hydrological
droughts changed' – and an answer of 'it's complicated' is cold comfort to water resource planners already
frustrated by the challenges of handling very large ensembles of future projections (i.e. deep uncertainty).
They may also question the finding of a lack of trends, given experiences with very extreme recent events





that, anecdotally, feel like 'something different' – 2018 and 2022 certainly are the kind of drought events we
expect to see more of in future, associated with high temperatures as well as rainfall deficits in the summer
half-year.

How then, should researchers, policymakers and water managers move forwards? We highlight here some
brief (and necessarily selective) recommendations for future research aimed at 'bridging the gap' between
observations and projections.

• Drought characterisation and 'types of drought'. Numerous authors have drawn distinctions between
'types' of UK drought, contrasting between within-year 'summer' droughts and long multiannual
droughts. Future studies should examine variability in different droughts, as in a warming world we
may expect differences between multiannual droughts (driven by successive dry winters) and short
duration droughts associated with increased evapotranspiration due to high temperatures. Given the
extreme aridity of recent droughts, analysis of 'flash' droughts assumes increasing importance. While
there are wide uncertainties in future projections of multiannual droughts (e.g. Watts et al. 2015),
future increases in summer half-year aridity are one of the more confident projections for the UK.
Noguera et al (2024) found limited evidence of increasing flash drought tendencies in meteorological
indices, but further analysis of the impact of recent flash droughts on hydrological systems, and how
this may change in future, warrants consideration, alongside multiannual droughts. Physically-based
storylines (Chan et al. 2022, 2023) are a promising avenue for appraising risk to given 'types' of event
and their combination.
• An even longer view of historical droughts although reconstructions have enriched our understanding
of past hydrological droughts, they still extend only to 1865 (CRU reconstructions) or 1890 (Historic
Droughts and MaRIUS reconstructions). Reconstructions have not been attempted, yet, for earlier
periods. This is an opportunity, given recent advances in extending meteorological datasets further
into the 19th century (Hawkins et al. 2022).  Monthly river flow reconstructions in Ireland have been
developed from 1766 (O' Connor et al. 2022), suggesting credible hydrological drought
reconstructions can be made over these very long time horizons. This would enable hydrological
comparisons with a growing body of knowledge on past meteorological droughts and their impacts
using either documentary sources (e.g. Pribyl & Cornes, 2020) or increasingly reliable paleoclimatic
reconstructions using dendrochronology (e.g. Loader et al. 2019).
• Improved understanding of climate drivers – going 'beyond the NAO'. In our review we highlighted
the barriers of using simple dipole-like atmospheric indices and recognised the emergence of process-
based studies looking at ocean-atmosphere dynamics on a hemispheric or global scale. Continued
improvement in our understanding of the drivers of drought on interannual to interdecadal timescales
can only help in our efforts to attribute emerging patterns of variability to anthropogenic or internally-



| 684 | driven factors – as well as to anticipate drought on seasonal to decadal timescales. While Section 4 |
| 685 | summarised the state of the art in tracking drivers of UK hydrological drought globally, a |
| 686 | comprehensive understanding of these long-distance influences on the North Atlantic is lacking, |
| 687 | highlighting the need for a coordinated effort to integrate research findings and form a complete |
| 688 | picture of the teleconnections of droughts. Greater integration between climate modelling simulations |
| 689 | and statistical hydrology will be pivotal and there is a role for new techniques such as using causal |
| 690 | inference approaches to quantify the teleconnection pathways (e.g., Kretschmer et al., 2021) or using |
| 691 | machine learning methods to the ascertain the impacts of the large-scale variability on water resources |
| 692 | (e.g., Kalu et al., 2023). |

- **Better discerning of the 'human factor' in drought.** The role of human interventions on river flows in
- general, and hydrological drought in particular, is a hot topic, academically, but also one that invites
- 'hot takes' – especially in the media and public narrative. Yet there is little evidence for a widespread
- footprint of human influences on changing hydrological drought patterns, despite the prevalence of
- demonstrable human impacts on river flow regimes. Improved attribution requires identification of
- both climate-driven and anthropogenic catchment changes, and quantifying their relative roles. This
- will require integration of field observation and climate and hydrological modelling, as well as further
- statistical and large-sample hydrological approaches. All these activities critically depend on
- observational datasets. While there have been efforts to improve the observational evidence base (e.g.
- the UK Benchmark network), major barriers remain – not least information on artificial influences
- and LULC change. Initiatives are underway to overcome these barriers, which will provide improved
- foundations for future studies. Improved datasets of human interventions and LULC open up the
- potential for large sample analyses based on AI methods that can isolate the role of climate factors
- and catchment factors, as demonstrated recently for flood trends by Slater et al. (2024).

- **'Bringing it all together' – better reconciliation of observations and models as a basis for decision-**
- **making.** Studies of observational trends have been calling for this since the mid-2000s (e.g.
- Hannaford and Marsh, 2006, Wilby , 2006, Wilby et al. 2008). The question is *'how?'* – because this
- is easier said than done given the relative brevity of hydrological drought records, and the
- aforementioned deep uncertainty of future projections. Increasingly, large ensembles of climate model
- or seasonal forecast model output are emerging as a powerful tool for contextualising flood and
- drought events (e.g. using the UNSEEN approach, applied to UK fluvial flood and hydrological
- drought events recently by Kay et al. 2024 and Chan et al. 2023). Such approaches allow us to look at
- 'worlds that might have been' – that is, seeing the observational time series as just one realisation of
- the past, and using large ensemble approaches to explore a much wider range of internal variability. In
- this context, some of the discrepancies seen between past trends and future projections (e.g. for the
- summer season) can be explained to a degree by random internal variability, and recent decades could
- have unfolded very differently. Deser and Phillips (2023) analysed climate trends using Single Model
- Initial condition Large Ensembles, or SMILES. Chan et al. (in preparation) has recently applied



similar approaches to hydrological drought variability in the UK to quantify signal-to-noise ratios and
time of emergence of drought trends.

Emerging analyses using such large ensemble and storyline approaches are a flexible, modular approach that
can be a unifying framework that enables decision-makers to explore each of these themes. They enable
exploration of past variability (including reconstructed droughts from centuries ago) alongside future
projections consistently, and one can explore risks and vulnerabilities to particular types of drought, including
extreme events that have not been sampled in observational records. Physically-based storyline approaches
have been used to explore the role of climate drivers in generating hydrological droughts (e.g. Chan et al.
2023, 2024) and, in principle, could also be used to help discern climate and catchment drivers – a
conceptually similar approach to disentangle climate and LULC trends was applied in Ireland by Harrigan et
al. (2014).  These approaches will be a cornerstone of future efforts to quantify variability in hydrological
drought. Seeing the past as only one realisation of many potential outcomes is an important shift in
perspective – one that poses important questions as to whether the observations of the recent past could create
a false sense of security. Future years and decades could increasingly see (worryingly) better agreement
between observations and projections.

In our introduction we argued a synthesis of research from the UK could provide a useful contribution to the
international debate on whether droughts have become more severe. However the story is complicated and
there is no 'smoking gun' of the influence of climate change on drought trends for the UK, nor any conclusive
evidence for worsening hydrological drought due to human activities. In fact the key finding is that there is in
fact little evidence to suggest any evidence towards worsening drought in the UK, alongside other studies that
suggest a similar picture across Northern Europe (Stahl et al. 2012; Pena Angulo et al. 2022) and other
temperate environments (Hodgkins et al. 2024). And yet, the picture of apparent discrepancies with near-
future projections is also shared elsewhere (e.g. in central Europe: Piniewski et al. 2021). The challenge of
providing straightforward assessments of observational change (for regional- to national-scale water managers
as well as global policy assessments like the IPCC) remains.

Nevertheless, our findings (and recommendations) resonate with experiences and insights from other settings
– there is much the UK can learn from the international community, and vice versa. For example: different
'types' of hydrological drought are routinely acknowledged and taxonomies have been produced (e.g. van
Loo, 2016 and references therein); there have been numerous efforts to reconstruct river flows over past
centuries (e.g. in France, Devers et al. 2024), suggesting pooling of approaches could be advantageous; the
subject of disentangling human and climate drivers has been the focus of dozens of papers (e.g. van Loon et
al. 2022) and our recommendations only underscore the importance of emerging approaches, whether data
science innovations (e.g. Slater et al. 2024) or socio-hydrological concepts (e.g. Ribiero-Neto et al. 2023) to



provide insights on the time evolution of UK droughts, coupling hydroclimatic, biophsysical and
socioeconomic drivers.


**7.  Code and data availability**
All river flow data used in this study is freely available on the UK National River Flow Archive:
https://nrfa.ceh.ac.uk/. The UK Benchmark Network is described in Harrigan et al. (2018) and a list available
at https://nrfa.ceh.ac.uk/benchmark-network. SSI calculated for observed river flows for the Low Flow
Benchmark Network and most NRFA catchments can be extracted from the UK Water Resources Portal
(https://eip.ceh.ac.uk/hydrology/water-resources/).

The reconstructed river flow data created by Smith et al (2019) and analysed by Barker et al. (2019) and here
is available in Smith et al. (2018): https://catalogue.ceh.ac.uk/documents/f710bed1-e564-47bf-b82c-
4c2a2fe2810e. The Standardized Streamflow Indices based on the reconstructions are available in Barker et
al. (2018): https://catalogue.ceh.ac.uk/documents/58ef13a9-539f-46e5-88ad-c89274191ff9.

NOAA's Extended Reconstructed SSTs, version 5 (Huang et al., 2017) is available at:
https://www.esrl.noaa.gov/psd/

The codes used in the extended analysis are available from the authors on request.

**Author contributions**
JH secured the funding, led the study and prepared the manuscript.  ST, AC and WC carried out extended
analysis and created the figures. SA commissioned the original review. All authors shaped the direction of the
review and contributed to the manuscript.

**Competing interests**
The contact author has declared that none of the authors has any competing interests

**Financial support**
The original version of this review was commissioned by the Environment Agency under award SC220020.
Additional funding to support the extended research and writing-up of this review was provided by (1) the UK
National Hydrological Monitoring Programme (supported by National Capability – UK, NE/Y006208/1), (2)
CANARI (NE/W004984/1) and (3) the Co-Centre for Climate + Biodiversity + Water Programme (grant no.
22/CC/11103) managed by Science Foundation Ireland (SFI), Northern Ireland's Department of Agriculture,
Environment and Rural Affairs (DAERA) and UK Research and Innovation (UKRI; grant NE/Y006496/1).



**Acknowledgments**
We acknowledge the Environment Agency for stimulating the original review, commissioned as part of a
series of studies reviewing the state of our knowledge on UK drought:
https://www.gov.uk/government/publications/review-of-the-research-and-scientific-understanding-of-drought
We thank the authors of other chapters, who provided feedback on earlier versions of the review.

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

**APPENDIX 1 – Methodology for extended analyses**



This section briefly describes the methods used in the extended analysis featured in this paper.

**1.  Trend analysis**

Annual values for all variables (Q50, Q70, Q90, and the four seasons) were firstly extracted for all NRFA stations meeting the record length criteria, and all Low Flows Benchmark Network stations (Harrigan et al. 2017). The $Qx$ variables are the exceedance flows that are very commonly used as flow regime metrics: Q50 is the river flow that is exceeded 50% of the time, Q70 70% of the time, and so on. Seasonal flows

The Standardised Streamflow Index accumulated over 3 months (SSI3) was calculated by fitting the Tweedie distribution to observed river flows of catchments in the LFBN. Comparing different probability distribution functions to fit river flow data for the purpose of calculating SSI, Svensson et al. (2017) concluded that the Tweedie distribution is most suitable for UK catchments. SSI fitted using the Tweedie distribution has previously been used for historical hydrological drought analyses in Barker et al. (2016; 2019) and to analyse future drought projections (e.g. Arnell et al. 2021). Hydrological drought characteristics were extracted from SSI3 following the method outlined in Table A1. SSI calculated for observed river flows for the Low Flow Benchmark Network and most NRFA catchments can be extracted from the UK Water Resources Portal (https://eip.ceh.ac.uk/hydrology/water-resources/).

**Table A1** Drought characteristics calculated from SSI3 for trend analysis.

| Drought characteristic | Method |
|---|---|
| Event | Consecutive periods of negative SSI3. Drought periods separated by one month are pooled to form the same event. |
| Drought duration | Annual total number of months in identified periods of drought conditions. |
| Max. intensity | Annual minimum SSI3 values within periods of identified droughts. |
| Mean deficit | Annual mean of SSI3 values within periods of identified droughts. |

The method for trend analysis was the standardised NRFA trend analysis toolkit described in Harrigan et al (2018a), which was based on established methods within hydrological literature. Monotonic trends were assessed using the Mann-Kendall (MK) test (Mann, 1945; Kendall, 1975), a non-parametric rank-based approach that is widely supported for use in streamflow analysis (e.g. Hannaford & Marsh, 2008; Murphy et


al, 2013). The magnitude of trends was estimated using the robust Thiel-Sen approach (Theil 1950; Sen 1968),
with trend magnitude expressed as a percentage change compared to the long-term mean (the Thiel-Sen
Average, TSA; Harrigan et al. 2018a).

The standardised MK test statistic (MKZs) follows the standard normal distribution with a mean of zero and a
variance of one. A positive (or negative) value of MKZs indicates an increasing (or decreasing) trend. The
probability of Type 1 errors set at the 5% significance level allowed the evaluation of statistical significance.
A two-tailed MK test was chosen, hence the null hypothesis of 'no trend present' (increasing or decreasing) is
rejected when MKZs is outside ±1.96 using traditional statistical testing.

The MK test requires data to be independent (i.e., free from serial correlation or temporal autocorrelation) as
positive serial correlation increases the likelihood of Type 1 errors or incorrect rejection of a true null
hypothesis (Kulkarni & von Storch 1995). All indicators were checked for positive lag-1 serial correlation at
the 5% level using the autocorrelation function (ACF) on detrended series. The linear trend used to detrend
the original time-series was estimated using the robust Theil–Sen estimator also used for characterising trend
magnitude.

Block bootstrapping (BBS) was used to overcome the presence of serial correlation and involves application
of the MKZs statistic to block resampled series that preserve any short-term autocorrelation structure.
Following guidance from Önöz & Bayazit (2012) regarding the optimal block length given the sample size
and magnitude of temporal autocorrelation coefficient, a block length of four years was chosen and applied
only when a series had statistically significant serial correlation – this occurred for 7,055 of the 231,245
single-station series analysed. In these cases, a robust estimate of the significance of the MKZs statistic was
generated from a distribution of 10,000 resamples where the null hypothesis of no trend is rejected when
MKZs calculated from original data are higher than the 9,750th largest (statistically significant increasing
trend) or lower than the 250th smallest (statistically significant decreasing trend) MKZs value from the
resampled distribution under a two-tailed test at the 5% level (Murphy et al. 2013).

**2.   Multitemporal analysis**

In addition to the fixed period trend analysis using a dense network of observed river flows in all NRFA
catchments, a multi-temporal trend analysis was also conducted following the methods set out in Hannaford et
al. (2013) using historical river flow and SSI reconstructions since 1891 (Barker et al. 2019) for nine example
catchments. Multi-temporal trend analyses are useful in providing additional context on the consistency of
trends over long multi-decadal timescales and help place short-term, fixed period trends in wider context. SSI
for the river flow reconstructions was calculated by fitting the river flow reconstructions using the Tweedie
distribution as described above. Hydrological drought characteristics were extracted from the SSI3 time series



for each catchment in the same approach as outlined in Table A1. The MK Z-statistic was calculated for each hydrological drought indicator and for every possible combination of start and end years over the entire river flow reconstruction period (1891-2015). A minimum window length of 27 years was chosen given the focus on interdecadal variability and the recognition that trend analyses are less robust and reliable for short time windows. SSI calculated from river flow reconstructions across the UK is available from the EIDC (https://doi.org/10.5285/58ef13a9-539f-46e5-88ad-c89274191ff9).

**3. Analysis of climate-streamflow relationships**

To identify remote teleconnections from large-scale climate drivers influencing UK droughts, we assess both concurrent and lagged relationships between the UK Standardized Streamflow Index (SSI) and global sea surface temperatures (SSTs). This approach accounts for long-term climate variability and helps establish robust relationships, aligning with the methodologies of Svensson and Hannaford (2019). This analysis is used to identify remote climate drivers, beyond the North Atlantic, that significantly influence UK droughts.

Our analysis utilizes observed catchment-scale SSIs at three-month accumulations from 850 catchments across the UK (Barker et al., 2022) and NOAA's Extended Reconstructed SSTs, version 5 (Huang et al., 2017).

Streamflow catchment characteristics in the UK vary regionally, so we applied k-means clustering on three-monthly accumulated SSI data to identify regions with similar streamflow patterns. Our analysis revealed three distinct regional clusters: the north-west UK, a transition zone, and the south-east UK (Figure 7a). This regional differentiation in SSI aligns with the streamflow clusters identified by Svensson and Hannaford (2019), where the north-west catchments are characterized by a fast response to rainfall, while the south-east catchments are groundwater-dominated, with delayed responses to rainfall.

We performed regressions of the area-averaged regional SSI time series for each of the three identified regions against the global SST dataset over the period of 1960 to 2020, evaluating both concurrent relationships (Figure 7b-d) and those with a six-month lag (Figure 7e-g) at each grid point.