# Peer review of "Have river flow droughts become more severe? A review of the"

_Hydrology and Earth System Sciences, 2024_

## Author Comment (AC1)

**REVIEWER 1**

Hannaford et al. use a large sample of stream gauges in the UK to assess whether hydrological drought properties and low flows have changed in recent decades as well as at longer time scales. They place the results of their analyses in the context of climatic and water/land use changes and suggest that evidence for worsening drought is minimal, especially amongst many confounding influences from human impacts. I commend the authors for the work they have presented here, but believe that substantial edits are required to clarify results as well as to more fully address the influence of human water and land use on drought properties and low flows. I believe that these edits will lead to a paper that will be of interest to the broad readership of HESS.

>>> Thank you for these positive comments and constructive suggestions on our extended review. We have responded to each below.

Comments and suggested edits bulleted below:

- Please check for consistency in author affiliation, postal code etc.

>>>We will address these address issues

- Please write out all abbreviations in figure captions. As written currently, it is hard for the reader to understand the content of the figures without referencing the text repeatedly.

>>>Thanks for this helpful observation and we agree there is a lack of clarity and user friendliness in places. We will review all captions.

- Figure 4 legend symbol up down is opposite of what is shown on the maps for intensity and deficit.

>>>Thanks, agreed, we changed the colour scheme and symbology to reflect consistency with previous figures (red/upward arrow = worsening, etc). The figure below is the revised version with the correct legend. Note for max. intensity and deficit, positive trends mean decreasing drought severity whereas for duration, positive trends mean increasing severity.

[Figure]

Color scale for figure 6 is the reverse of intuition or the statistic seems reversed. Prior trend plots showed drought properties becoming longer and more intense through time in the south and shorter and less intense in the north.

>> the colours are correct. we will amend in the legend to emphasize again that positive trends (blue) indicate amelioration of drought and negative trends (red) indicate increasing severity. The question about the south is an interesting one, partly reflecting the fact there are very few catchments in the southeast, only the Lambourn, in the set used by Barker et al. 2019. For balance, we could add three more sites to give more in this area and a better distribution overall. The figure below shows multi-temporal trends re-calculated using SSI-12 for nine sites, including three new sites in East Anglia (Stringside), Kent (Bull) and southwest England (Otter), for example (we will review in the revised version).

[Figure]

- Current figure 6 with the legend interpreted as shown implied that for trends tests starting and ending in more recent years, accumulated drought deficit has increased significantly for several sites in the north. Cree, Allen, Ellen as examples.

>>>>Legend is correct. As above, positive trends (blue) indicate amelioration of drought and negative trends (red) indicate increasing severity

- There are several studies that I believe should be cited in the introduction and discussion to more thoroughly place the present study in the context of other recent studies on hydrological drought patterns and trends:

  o Brunner, M. I., Swain, D. L., Gilleland, E., & Wood, A. W. (2021). Increasing importance of temperature as a contributor to the

spatial extent of streamflow drought. *Environmental Research Letters*, *16*(2), 024038.

- o Dudley, R. W., Hirsch, R. M., Archfield, S. A., Blum, A. G., & Renard, B. (2020). Low streamflow trends at human-impacted and reference basins in the United States. *Journal of Hydrology*, *580*, 124254.

- o Hammond, J. C., Simeone, C., Hecht, J. S., Hodgkins, G. A., Lombard, M., McCabe, G., ... & Price, A. N. (2022). Going beyond low flows: Streamflow drought deficit and duration illuminate distinct spatiotemporal drought patterns and trends in the US during the last century. *Water Resources Research*, *58*(9), e2022WR031930.

- o Konapala, G., & Mishra, A. (2020). Quantifying climate and catchment control on hydrological drought in the continental United States. *Water Resources Research*, *56*(1), e2018WR024620.Please capitalize the first word of figure caption in every figure caption.

- o Tijdeman, E., Barker, L. J., Svoboda, M. D., & Stahl, K. (2018). Natural and human influences on the link between meteorological and hydrological drought indices for a large set of catchments in the contiguous United States. *Water Resources Research*, *54*(9), 6005-6023.

- o Van Loon, A. F., & Laaha, G. J. J. O. H. (2015). Hydrological drought severity explained by climate and catchment characteristics. *Journal of hydrology*, *526*, 3-14.

>>>>Thanks for these suggestions, we will review and consider addition. Although, to be honest, our review is already long and we have given some consideration of the international literature but we cannot realistically cover the wider issues fully – there are already other far broader (geographically) and more comprehensive (thematically) reviews out there. For example, I note there is a high number of papers suggested here from the US, and while I agree that there has been some excellent work on trends in drought/low flows there, arguably we should consider many other parts of the world also.

We will review and add papers where we feel there is a relevance and benefit from the topics covered/findings, as appropriate to our study.

- Figure 8 what does s in figure caption stand for? What is BFIHOST?

>>>Apologies, we took the caption from the original and modified but did not spot everything that needed explanation. We will add description of the FAR codes (S = sewage treatment works) and also BFIHOST (Baseflow Index as estimated from the HOST classification). We will also add refs and links (e.g. https://nrfa.ceh.ac.uk/feh-catchment-descriptors)

- Considering the extensive research and monitoring program that the authors are using for this research, I find the analysis and discussion of human impacts on drought and low flows to be incomplete. Are there not spatial datasets that report on the changing patterns of water withdrawals, impoundments, land use changes that would make it possible to more fully assess how trends in streamflow signatures have been impacted by human flow regulation, land use change, and groundwater regulation for individual watersheds as well as the aggregate response across regions? This would substantially extend the impact of this paper.

>>> This is an insightful and useful comment. Alas, I fear a 'discussion of human impacts on drought and low flows' will always be incomplete! The short answer is: no, there are not really appropriate datasets to allow this and instead the last decade or so has seen many large-sample analyses that have made inroads into the issue, but we do lack the kind of datasets to fully unpack this.

There have been some significant advances in some new datasets that will allow this kind of analysis – but published studies using them are in their infancy. We had hoped this would come across in the 'story' of this section but I appreciate that it is perhaps a bit lost in the narrative, especially for international audiences (based as it was on a UK-focused review that made certain assumptions about existing knowledge). I agree with the sentiment of the reviewer that this could be much clearer and could flow better to guide the reader – we will review this whole section and make our arguments clearer as to what the data situation is, what science has been done, what the limitations are and what the next steps are.

- Tabular summaries of the fraction of all sites and fraction of least disturbed sites with significant trends in low flow and drought metrics are needed. Preferably at least splitting into northern and southern UK regions, as well as into classes of land use / land cover (e.g. agricultural, forested, urban, heavily regulated by dams). These tables would help to synthesize the main results displayed in the figures in a way that makes digesting these results easier for the reader, and could also enhance the analysis of human impact on hydrological drought properties and low flows in the UK. See Dudley et al. (2020) as an example:
  - Dudley, R. W., Hirsch, R. M., Archfield, S. A., Blum, A. G., & Renard, B. (2020). Low streamflow trends at human-impacted and

reference basins in the United States. *Journal of Hydrology*, *580*, 124254.

>>>I agree that this is a good idea, and thanks for the suggestion of the paper as inspiration – I agree it is a great exemplar. Although I feel that it is somewhat beyond the scope of a review/extended review to do this – it would be a reasonably substantial new analysis. We can add tables of trend summaries for each region (as in Hannaford et al. 2021, in the ref list), and by all/benchmark as a first pass effort to look at near-natural/influenced.

We will consider the option of looking at LULC/influence, but I think this is beyond scope. It would be great to emulate the Dudley et al. 2020 study but I think it would need much more analysis  - to look at meaningful stratification of some of the LULC variables, say, given the different sampling of catchments across them. There are precedents for looking at the impact of static catchment characteristics on dynamic streamflow catchments, and one problem is really picking apart causality given the co-varying of many different landscape properties across the NW-SE Gradient of the UK – the wet, upland, mountainous and resistant geology of the NW versus dry, lowland, baseflow-dominated catchments of the SE. As illustrated by Chiverton et al. 2015 – where LULC properties (e.g. % arable land) were shown to be drivers of temporal variability, but in fact the LULC properties are largely just a function of arable land being in dry, low-lying settings in eastern England (where there are major aquifers). We will add some commentary on this.

A great suggestion, but one for a follow-up study, in my opinion - would definitely much more space and many more figures than we can spare in an extended review.

**REVIEWER 2**

**Summary**

This paper presents a review and extended analysis of streamflow drought in the UK based on a dense network of observed river discharges across all NRFA catchments. First, the study analyzes changes in drought severity across the UK using observed data, followed by an extended analysis with reconstructed data. Second, the paper explores key drivers of streamflow drought, focusing on climatic and human influences. The authors found little evidence that drought will become more severe, which contradicts near-future climate projections and anticipated human disturbances. Furthermore, they also highlighted some recommendations for researchers, policy makers, and water managers on moving forward.

**Assessment**

This paper presents new evidence that challenges the notion of worsening drought over recent decades due to climate change. The analysis is based on the in-situ observational data, which provides a reliable basis for the findings. However, while these findings may be specific to the UK, and not directly applicable to many regions (e.g., southern Europe), the authors argue for their broader relevance. The manuscript is interesting and well written. I have a few minor comments below and three general comments, but only for clarification and improvement. I believe this work is well-suited for publication in HESS.

>>>Thank you, Samuel, for your very positive comments on the review and its wider transferability.

**General Comments**

I have three general comments regarding the manuscript, all aimed at clarification, suggestion, and improvement.

1. As mentioned above, the findings of this study may be specific to the UK. Numerous studies have highlighted that drought will be more severe in southern European regions. I suggest that the authors reconsider their statement about the broader relevance of their findings or specify which aspects of the findings may have broader applicability.

>>>Thanks for this. Along with the comment from R1 too, this is helpful, we agree that we should further contextualise our work with the international literature. We will revisit this in the intro and the discussion, and will be sure to add material on the wider European domain where the lack of trends/trends towards drought amelioration in the NW are countered by stronger changes in S and E Europe. (n.b. In the original review there was a whole section on European trends that was omitted due to space, we will incorporate some of this).

2. I recommend incorporating more quantitative results instead of qualitative descriptions. The authors tend to present their results in a qualitative way e.g., only mentioning increase or decrease without providing precise percentage changes in trend magnitude (see Fig. 2).

>>>This is really in keeping with the 'extended review' nature of our article. However, we will add some quantitative headlines of % of sites with given changes, for example (in keeping also with reply to reviewer 1's final comment). We emphasise that this will be brief and high-level given the scope and purpose of our review.

3. There is inconsistency in the presentation of result, especially for Figure 5. The authors used SSI-3 in Figure 4, SSI-12 in Figure 5, and then return to SSI-3 in Figures 6 and 7. If possible, I suggest to replace SSI-12 in Figure 5 with SSI-3, as the author of Figure 5 is also a co-author of this paper.

>>> This is a good point, and such a figure exists (in SI of Barker et al. 2019, below). Nevertheless, SSS12 is a more useful descriptor of the main historical droughts, especially in the flashier north of the country. I propose we keep Fig 5 as SSI12 and change Fig 6 to SSI12 but retain the alternatives as a new SI in this paper.

[Figure]

Figure S2: Heatmap of SSI-3 for LFBN catchments (arranged roughly from north to south on the y-axis with one row per catchment and regions marked for clarity) from 1891 to 2015.

**Line by line comments**

L refers to line and P refers to page.

**P2L48-56**: I suggest moving the second paragraph to the end of introduction. This paragraph presents the objective of the study, and in my opinion, it is disturbing the flow of introduction.

>>>We will do this

**P3L80**: Maybe place a comma before "it is necessary to quantify....."

>>>OK

**P3L85**: What do the authors mean with "international standard"?

>>>We just mean relative to international norms. We should provide evidence and a reference to support this

**P4L122**: I am wondering how we can define accumulation deficit for SSI? From my understanding, accumulation deficit can be applied only for threshold approach.

>>>Accumulated deficit is widely used with SSI, as in the previously cited papers (and many other papers that use the SPI. It is perhaps more abstract a quantity in SSI units, but the concept is just the same as with the threshold level approach. (As an aside, I would argue the accumulated deficit is not always so meaningful for the threshold level method either unless rendered into a volume of 'lost' water, but this is rarely done and it is often presented in terms of cumecs (flow rates) which is similarly hard to visualise/analyse.

**P12L324**: Maybe place a comma before "there is generally a contrast….."

>>OK

**P15L392**: Please provide example of literatures/references when you say many literatures.

>>Agreed, we will cite some introductory refs as entry points

**P15L398**: Write the abbreviation of East Atlantic "(EA)" here since it is the first time EA is mentioned.

>>OK

**P15L415**: What is SCA?

>>Scandinavia pattern, we will spell out.

**P20**: Figure 8. In the figure caption, please use letter a and b instead of top and bottom. Also, please explain what N, G, S, GS, and others are.

>>Agreed, we will amend the caption (see also reply to R1) and labeling

**P20L569**: Here the authors write "It follows that…." As opening paragraph. What "it" refers to?

>>We mean the whole previous section, we will be clearer

**P21L587**: Maybe place a comma before "LULC have been very….."

>>OK

**P22L629**: Maybe place a comma before "it is important not to….."

>>OK

**P22L645**: What do the authors mean with "cold comfort"? Also maybe write "….who are already frustrated…."

>>We have used some more colloquial English in places to enhance readability/style as this is a review. We agree that maybe this could be confusing for international readers (although in my opinion there is a  balance here and often scientific journals benefit from some added 'everyday' colour [add examples?].

 **P25L740**: I think you should reduce the jargon "smoking gun"

>>as above. Though to my mind 'smoking gun' idiom is very widely used and commonly understood in our field due to climate change connotations. We will review this whole section for style.

**P25L741**: Put comma and remove second "in fact" -> In fact, the key finding is that there is little evidence….

>>OK

**P36L1125**: It seems that the sentence is not finish. "Seasonal flow"

>>Thanks for spotting this, we will amend!